# Evaluation of the acceptability in France of the vaccine against papillomavirus (HPV) among middle and high school students and their parents

Jean-François Huon[1]*, Antoine Grégoire[2], Anita Meireles[2], Maëva Lefebvre[2,3], Morgane Péré[4], Julie Coutherut[2], Charlotte Biron[2,3], François Raffi[3], Valérie Briend-Godet[2]

1 Clinical Pharmacy of the UHC of Nantes, and UMR INSERM 1246 SPHERE Universities of Nantes and Tours, Nantes, France, 2 Center for the Prevention of Infectious and Transmitted Diseases of the UHC of Nantes, Nantes, France, 3 Department of Infectious Diseases, and CIC 1413, INSERM, University Hospital Center of Nantes, Nantes, France, 4 Plateforme de Méthodologie et Biostatistique, Direction de la Recherche, CHU de Nantes, Nantes, France

* jeanfrancois.huon@chu-nantes.fr

**Data Availability Statement:** Dataset is available at https://doi.org/10.17026/dans-x52-g2qu.

## Abstract

### Background

The pathogenic and oncogenic roles of papillomavirus (HPV) infections have been documented and shown to occur in women as well as in men. While other countries have already extended their vaccination guidelines to include boys, in 2019 the French National Authority for Health validated implementation of HPV vaccination in the 2020 vaccination schedule. There is, however, a climate of distrust in regard to vaccination in France, and there have been few studies to date regarding the acceptability of HPV vaccination in boys in France. The aim of this study was, therefore, to evaluate the acceptability of extending the recommendations for HPV vaccination in men, among middle and high school students and their parents.

### Methods

Our study (HPVac) was a prospective, multicenter, departmental, and descriptive survey applied to a sample of male middle and high school students attending schools in the Loire-Atlantique department and their parents. It took place from January 2017 to January 2018.

### Results

We analyzed the information obtained from 127 parent questionnaires and 145 children questionnaires. In terms of acceptability, 36.6% (n = 53) of the children and 37.8% (n = 48) of the parents were in favour of being vaccinated or of having their children vaccinated against HPV (51.7% (n = 75) and 50.4% (n = 64), respectively, were undecided). The perception of a risk stemming from HPV infection was positively associated with acceptability of the HPV vaccine. Being against vaccines in general, being discouraged by their parents, parents thinking that their child is not at risk, and the belief that the vaccine is not mandatory were arguments cited and significantly associated with a willingness to be vaccinated.

**Funding:** The author(s) received no specific funding for this work.

**Competing interests:** The authors have declared that no competing interests exist.

## Conclusion

This study revealed a lack of information among boys and their parents about HPV and its vaccination. It also clearly showed that taking time to discuss the consequences of an infection and the merits of being vaccinated can help parents overcome their reluctance. The children then generally go along with their parent's choice.

## Introduction

Infection by human papillomavirus (HPV) is one of the three main sexually transmitted infections (STIs) involving the general population, and it is the most common viral STI [1–4].

The pathogenic and oncogenic role of HPV infections has been documented and shown to occur both in women and in men (condylomas, and cancers of the cervix, the vulva, the anus, and the oropharynx, as well as of the penis) [5, 6].

Vaccination against HPV has been shown to be effective in real-life situations, as it decreases the incidence of infections by papillomavirus (types 6, 11, 16, and 18) as well as the occurrence of condylomas, and of low- and high-grade lesions of the cervix [7].

Vaccination against HPV has been recommended in France since 2007 in girls of 11 to 19 years of age, with the aim of reducing the incidence of cervical cancer, [8, 9]. Gardasil 9®, available in France since 2018 and now the only entity recommended for this indication, has also received marketing approval for the prevention of condylomas due to specific types of HPV (6 and 11) as well as the prevention of precancerous and cancerous anal lesions.

Increased knowledge regarding the involvement of HPV in the genesis of other cancers affecting men led the French High Council of Public Health (HAS) to issue an opinion in February of 2016 on the vaccine strategy in boys. In addition to the recommendation to extend the vaccination to individuals of up to 26 years of age for men who have sexual relationships with other men, it concluded that the cost-efficacy ratio of universal vaccination becomes favourable when all of the pathologies linked to HPV are considered and/or subject to elevated vaccination coverage in boys when the vaccination coverage in girls is low (< 40%, as is the case in France) [10]. This has led the HAS to recommend extending HPV vaccination to boys of 11 to 14 years of age for the vaccination schedule in 2020.

Other countries have already extended their vaccination guidelines to include boys. For example, 15 countries in Europe have included the vaccination of boys in their immunization schedules (Austria, Belgium, Croatia, the Czech Republic, Denmark, Germany, Ireland, Italy, Liechtenstein, Luxembourg, Norway, Slovakia, Sweden, Switzerland, and the United Kingdom) [11, 12].

While a number of countries, such as the US have studied the acceptability of HPV vaccination in boys [13, 14], there have been few studies to date regarding the acceptability of HPV vaccination in boys in France. Furthermore, it is difficult to extrapolate foreign data to the French context, particularly as there is a climate of distrust in regard to vaccination in France [15]. Fear of side effects, misinformation spread by anti-vaccine lobbying, and poor knowledge of vaccine-preventable diseases is exacerbated by a loss of confidence in institutions. Indeed, there has been a succession of health scandals (or that are perceived as such) in recent years in France. Among these, the controversy surrounding the side effects of the hepatitis B vaccine, which led to the suspension of the school vaccination campaign at the end of 1998, could explain this lack of confidence [16].

The main objective of this study was, therefore, to evaluate the acceptability of extending the recommendations for HPV vaccination in men, among middle and high school students and their parents, and hence to better assess the feasibility of generalization of HPV vaccination for boys. The secondary objectives were evaluation of the knowledge of those surveyed regarding HPV and the vaccine, their sources of information, as well as factors impeding or promoting vaccination against HPV.

## Materials and methods

Our study (HPVac) was a prospective, multicentre departmental, non-controlled, quantitative, and descriptive survey applied to a sample of grade 4 and grade 3 male middle and high school students attending schools in the Loire-Atlantique department (HPVac_child) and their parents (HPVac_parent).

The study was carried in collaboration with the Medical Affairs and Research Board and the Medical Evaluation and Epidemiology Unit of the University Hospital Centre of Nantes.

The facilities were randomly selected among all of the middle and high schools in the Loire-Atlantique department, whether public or private, general or vocational. Each facility was assigned a numerical code to allow the questionnaires to be tracked (Period 1). In light of the low rate of participation in the study, it was secondarily decided to propose participation in the study to all of the scholastic facilities in the Loire-Atlantique department (Period 2). No incentives were offered to motivate the respondents to complete the questionnaire.

Among the facilities that accepted to participate in the study, the parents of male pupils (minors or adults) enrolled at a middle school (in grade 4 or grade 3) or high schools of the Loire-Atlantic had to agree to participate in the research and provide their consent for the participation of their child(ren). The children in question also had to provide their consent. The other criteria for inclusion were being a French-speaker and having access to the internet.

The study received a favourable assessment from the Nantes Ethics in the Area of Health Group (GNEDS). The consent of each participant was collected after they had been provided an information sheet. Each participant had remote access to the questionnaire through a link to a website in the letter that was sent. No nominative data was collected, thus ensuring anonymity and confidentiality of the data.

The study took place from January 2017 to January 2018. The participants could access the online questionnaire through the website link provided in the letter that was sent (Period 1) and in the flyer that was distributed to the scholastic facilities (Period 2).

The questionnaires comprised 5 parts (Appendix 1): socio-economic data; the perception of vaccination in general; knowledge of HPV and the vaccine against HPV; parental acceptance of the vaccine against HPV; sexuality, partners, and at-risk behaviours.

The questions regarding sexual practices and sexual orientation were excluded voluntarily so as to obtain approval from the authorities and to increase adhesion by the participants. Some of the questions were specific for either the children or the parents, the others applied to both populations. The questionnaires were tested on fifteen pairs of parents so as to evaluate the readability and the comprehension of the questionnaires by the targeted individuals. This allowed evaluation of the relevance of the questions and possible formulation biases, and it also ensured that the questionnaire was suitable for the intended analyses. This stage of the test also led to the exclusion of grade 6 and grade 5 middle school pupils. The degree of maturity of this category was deemed to not be compatible with the content of the questionnaire, as sexuality is addressed in middle schools starting in grade 4. The parents of these children were, therefore, not solicited. Finally, some questions were deleted due to misunderstanding or duplication. This was the case for questions regarding the notion of increased risk in case of

vaccination, the importance of vaccinating partners, or -when the boy respondent had a sister- her vaccination status.

All of the analyses of the data were carried out using R software version 3.4.3 for all of the analyses. The qualitative variables are described as the numbers and percentages of the modalities, and the quantitative variables as the minimum, maximum, mean, standard deviation, and median. The concordance of the parent couple/child in regard to acceptance of HPV vaccination was evaluated using Cohen's kappa test. Associations between the socio demographic characteristics and the adherence to vaccination were probed on an exploratory basis using univariate and multivariate (forward method) logistic regression models. For qualitative variables with too low a number, a Fisher's exact test was used. The applied confidence interval was 95%. The significance level was set at 5%.

## Results

Over the course of Period 1, 36 scholastic facilities (middle and high schools) were asked to participate in the survey and 212 were solicited over the course of Period 2, thus amounting to a total of 248 facilities that were asked to participate. Details of the inclusions per period and according to the scholastic facility are presented in Fig 1. The overall rate of participation by the scholastic facilities was 11.7% (n = 29): 6 public middle schools, 5 private middle schools, 14 public high schools, and 4 private high schools.

Over the course of Period 1 and Period 2, 3856 letters and 5405 flyers, respectively, were distributed to the scholastic facilities that had agreed to participate in the survey, amounting to a total of 9,261 families that were solicited. The questionnaire was completed by 128 parents and 146 children. After being scrutinized for errors, a single questionnaire from each category was excluded due to aberrant data. Ultimately, 127 parent questionnaires and 145 children questionnaires were obtained for the analysis (a rate of participation of 1.37% and 1.6%, respectively).

The sociodemographic characteristics of the parents are summarized in Table 1. In terms of the children, 34.5% were in middle school (n = 50) and 65.5% were in high school (n = 95). The median age of the children was 16 years [13–18].

All up, 73.8% (n = 107) of the children and 70% (n = 89) of the parents surveyed were generally in favour of vaccination. Furthermore, 86.2% (n = 125) of the children and 73.2% (n = 93) of the parents had confidence in vaccines in general. Hepatitis B, Influenza, and HPV vaccines were the three vaccines for which both groups expressed the most distrust (50%, 26.5%, and 29.4%, respectively, for the parents and 15%, 20%, and 10%, respectively, for the children). Involvement of the parents in the decision was reported by 91% of the children (n = 132), while only 83.5% (n = 106) of the parents reported intervening in the decision to vaccinate their children.

The data regarding the knowledge of HPV and its vaccine are summarized in Table 2 for the surveyed individuals: 85% of the parents (n = 108) and 84.8% of the children (n = 123) reported not being sufficiently informed. Of these, 76.9% of the parents (n = 83) and 70.7% of the children (n = 87) wished to receive more information on HPV and the vaccine.

The treating physicians and the school doctor were the two most often cited sources of information: 65.1% (n = 54) and 59% (n = 49), respectively, of the parents and 50.6% (n = 44) and 55.2% (n = 48), respectively, of the children. The data regarding the acceptability of the vaccine are summarized in Table 3.

The factors positively associated with acceptability or refusal of the vaccine among the children are summarized in Table 4. The probability of being amenable to being vaccinated was significantly higher when the children were worried about the consequences of being infected,

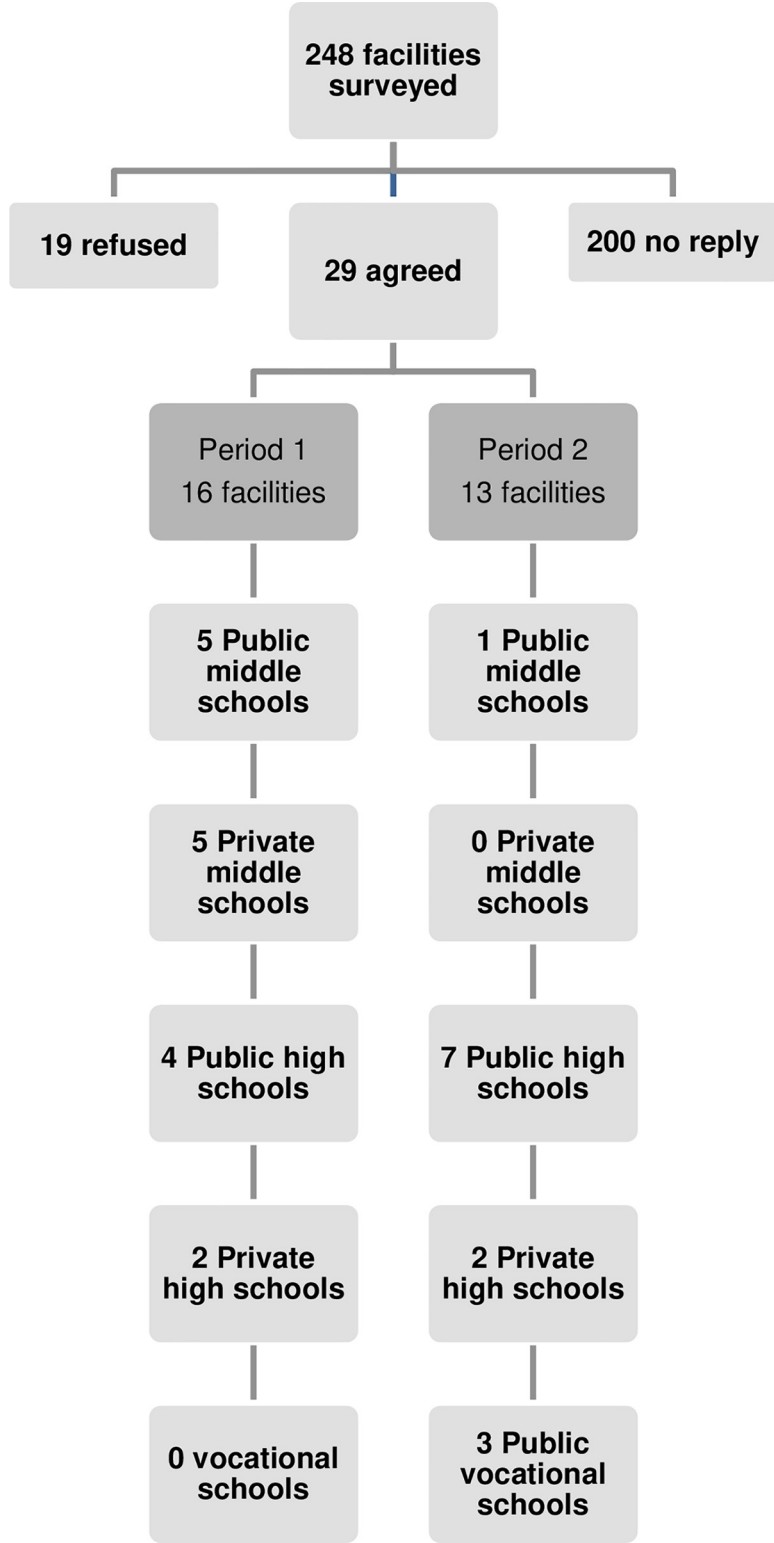

**Fig 1. Diagram of the facilities.**

**Table 1. The sociodemographic characteristics of the surveyed parents.**

| Age of the parents/guardians (years) | Median [min-max] | |
|---|---|---|
| Age of the father | 48 [35–60] | |
| Age of the mother | 46 [34–57] | |
| Age of the legal guardian | 44 [18–56] | |
| **Variables (number)** | **(%)** | **Denominator** |
| **Number of children, average** | 1.2 | |
| **Socioprofessional category of the responding parent** | | 127 |
| Agricultural worker, farmer | 1 (0.8) | |
| Craftsperson and head of a company | 5 (3.9) | |
| Employee | 53 (41.7) | |
| Intermediate professions | 8 (6.3) | |
| White-collar and gray-collar workers | 39 (30.7) | |
| Laborer | 14 (11) | |
| Retirees | 0 (0) | |
| Individuals not in the workforce | 7 (5.5) | |
| **Socioprofessional category of the partner** | | 127 |
| Agricultural worker, farmer | 1 (0.8) | |
| Craftsperson and head of a company | 6 (4.7) | |
| Employee | 45 (35.4) | |
| Intermediate professions | 13 (10.2) | |
| White-collar and gray-collar workers | 41 (32.3) | |
| Laborer | 14 (11) | |
| Retirees | 0 (0) | |
| Individuals not in the workforce | 7 (5.5) | |
| **Geographical distribution** | | 127 |
| Town | 76 (59.8) | |
| Rural | 51 (40.2) | |
| **Is there a general practitioner in your town?** | | 127 |
| Yes | 117 (92.1) | |
| No | 10 (7.9) | |

and perceived that there was a risk of HPV infection, that the vaccine could be effective and protect against serious diseases. The latter result was confirmed by the multivariate analysis (OR = 16.50 [1.61; 169.03]; p = 0.0182). The probability of refusal was higher when the children were discouraged by their parents, this being confirmed by the multivariate analysis (OR = 35.09 [3.59; 343.33]; p = 0.0022).

A lack of knowledge regarding the vaccine, the cost of the vaccine, and fear of secondary effects, while often cited, was not significantly associated with the desire to not be vaccinated. Nor was there a significant association between the age of the children (p = 0.412), attending middle or high school (p = 0.176), being at a private or public facility (p = 0.891), the socioprofessional category of the parents (p = 0.105), living in an urban versus a rural area (p = 0.298), and the presence of a general practitioner in the town where they resided (p = 0.184), and the children's favourable attitude toward HPV vaccination.

In terms of the parents, being opposed to vaccination in general, the notion that one's child is at risk, and the fact that the vaccine is not compulsory were the arguments significantly associated with the desire to not have one's son(s) vaccinated. A lack of knowledge regarding the vaccine, fear of the secondary effects, and the fact that the subject was not raised by the treating doctor were the most often cited reasons, although they were not significant.

**Table 2. Knowledge regarding papillomavirus and its vaccine.**

| | Children (N = 145) | Parents (N = 127) |
|---|---|---|
| **What is the mode of transmission of human papillomavirus?** | | |
| Sexual relations | 75.9% (n = 110) | 79.5% (n = 101) |
| Blood | 24.1% (n = 35) | 20.5% (n = 26) |
| Saliva | 10.3% (n = 15) | 10.2% (n = 13) |
| Air | 0.7% (n = 1) | 0.8% (n = 1) |
| Others | 2.8% (n = 5) | 1.3% (n = 2) |
| Don't know | 22.10% (n = 32) | 19.7% (n = 25) |
| **Does use of a condom protect against papillomavirus?** | | |
| Yes | 59.3% (n = 86) | 60.6% (n = 77) |
| No | 11% (n = 16) | 11.8% (n = 15) |
| Don't know | 29.7% (n = 43) | 27.6% (n = 35) |
| **Have you heard of the vaccine against papillomavirus?**[*] | | |
| Yes, on several occasions | 11% (n = 16) | 41.7% (n = 53) |
| Yes, once | 31% (n = 45) | 20.5% (n = 26) |
| No | 58% (n = 84) | 37.8% (n = 48) |
| **What does the vaccine protect against?** | | |
| Cervical cancer | 53.8% (n = 78) | 71.7% (n = 91) |
| Breast cancer | 3.4% (n = 5) | 2.4% (n = 3) |
| Other genital cancers | 21.4% (n = 31) | 22.8% (n = 29) |
| Other cancers (ENT. . .) | 6.2% (n = 9) | 8.7% (n = 11) |
| Genital warts | 14.5% (n = 21) | 18.1% (n = 23) |
| AIDS | 6.9% (n = 10) | 3.9% (n = 5) |
| Other STIs | 14.5% (n = 21) | 11% (n = 14) |
| Others | 1.4% (n = 1) | 1.6% (n = 2) |
| Don't know | 29.7% (n = 43) | 16.5% (n = 21) |

[*]The main means by which the participants had heard about the vaccine against HPV were: the media (62%, n = 49) and the doctor (31.6%, n = 25) for the parents (N = 79) and a teacher at the facility (47.5%, n = 29) and the school doctor (34.4%, n = 21) for the children (N = 61).

**Table 3. Data regarding the acceptability of the vaccine.**

| | Children (N = 145) | Parents (N = 127) |
|---|---|---|
| **Are you in favor of vaccination against HPV of young girls?** | | |
| Yes | 60.7% (n = 88) | 59.1% (n = 75) |
| No | 2.1% (n = 3) | 7.9% (n = 10) |
| Don't know | 37.2% (n = 54) | 33% (n = 42) |
| **Are in favor of vaccination against HPV in boys aged between 11 and 26 years?** | | |
| Yes | 57.9% (n = 84) | 48.8% (n = 62) |
| No | 4.8% (n = 7) | 6.3% (n = 8) |
| Don't know | 37.2% (n = 54) | 44.9% (n = 57) |
| **Would you like yourself or your son to be vaccinated?** | | |
| Yes | 36.6% (n = 53) | 37.8% (n = 48) |
| No | 11.7% (n = 17) | 11.8% (n = 15) |
| Don't know | 51.7% (n = 75) | 50.4% (n = 64) |

**Table 4. Factors associated with acceptability or refusal of the vaccine among children and their parents.**

| Variables | Children | | | | Parents | | | |
|---|---|---|---|---|---|---|---|---|
| | N | OR | IC 95% | p-value | N | OR | IC 95% | p-value |
| | Arguments for getting vaccinated | | | | Arguments that would incite to vaccine his son | | | |
| There is a risk associated with HPV infection | 70 | 9.78 | [2.03; 47.13] | 0.0045 | 63 | 28.00 | [3.38; 232.26] | 0.0020 |
| This vaccine protects against serious diseases | 70 | 14.58 | [3.00; 70.88] | 0.0009 | 63 | 2.75 | [0.77; 9.86] | 0.1204 |
| This vaccine is effective | 70 | 4.49 | [1.16; 17.47] | 0.0301 | 63 | 2.53 | [0.71; 9.07] | 0.1542 |
| This vaccine protects against genital warts | 70 | * | - | 0.007 | 63 | * | - | 0,052 |
| This vaccine is reimbursed (65%) | 70 | 1.74 | [0.34; 8.88] | 0.5030 | 63 | * | - | 0.013 |
| This vaccine has few or no side effects | 70 | 0.63 | [0.18; 2.16] | 0.4616 | 63 | 0.33 | [0.10; 1.08] | 0.0659 |
| I am worried about the consequences of being infected | 70 | 14.29 | [1.77; 115.62] | 0.0127 | 63 | 4.26 | [0.86; 21.02] | 0.0754 |
| My family and friends recommend it to me | 70 | 0.43 | [0.12; 1.54] | 0.1943 | 63 | * | - | 0,998 |
| My doctor advises me to | 70 | 1.14 | [0.21; 6.10] | 0.8772 | 63 | 2.00 | [0.22; 18.08] | 0.5372 |
| No argument for | 70 | 0.03 | [0.00; 0.25] | 0.0014 | 63 | * | - | - |
| | Arguments that would not encourage children to be vaccinated | | | | Arguments that would not encourage to vaccine his son | | | |
| I'm against vaccines in general | 70 | * | - | <0.001 | 63 | * | - | < 0.001 |
| This vaccine is not mandatory | 70 | 1.79 | [0.51; 6.25] | 0.3604 | 63 | 5.12 | [1.41; 18.67] | 0.0132 |
| This vaccine is not effective | 70 | 0.78 | [0.22; 2.78] | 0.7007 | 63 | 1.22 | [0.32; 4.61] | 0.7662 |
| I don't know enough about the vaccine | 70 | 1.88 | [0.62; 5.73] | 0.2654 | 63 | 2.78 | [0.84; 9.12] | 0.0927 |
| This vaccine is too expensive | 70 | 0.98 | [0.31; 3.06] | 0.9669 | 63 | 0.43 | [0.12; 1.54] | 0.1951 |
| My parents don't recommend it | 70 | 8.57 | [2.11; 34.91] | 0.0027 | NC | NC | NC | NC |
| My friends and family don't recommend it | NC | NC | NC | NC | 63 | 3.54 | [0.45; 27.61] | 0.2280 |
| I am not worried about the consequences of being infected | 70 | 3.57 | [0.65; 19.68] | 0.1437 | 63 | 8.36 | [1.35; 51.64] | 0.0222 |
| I'm afraid of the side effects | 70 | 0.98 | [0.31; 3.06] | 0.9669 | 63 | 3.00 | [0.91; 9.91] | 0.0715 |
| I'm afraid of needles | 70 | 4.00 | [0.99; 16.09] | 0.0509 | NC | NC | NC | NC |
| The number of injections | NC | NC | NC | NC | 63 | 1.38 | [0.36; 5.28] | 0.6361 |
| My child is not at risk | NC | NC | NC | NC | 63 | 0.09 | [0.01; 0.89] | 0.0399 |
| My doctor didn't tell me about it | 70 | 0.73 | [0.18; 2.98] | 0.6633 | 63 | 1.09 | [0.29; 4.07] | 0.8970 |
| My doctor doesn't recommend it | 70 | 0.82 | [0.20; 3.36] | 0.7807 | 63 | 0.58 | [0.11; 3.02] | 0.5221 |
| No argument against | 70 | 0.31 | [0.04; 2.61] | 0.2784 | 63 | * | - | - |

* For qualitative variables with too low a number, a Fisher's exact test was used.

Of the parents who stated that they had a daughter who was 11 years of age (or 46.5% of the parents, n = 59), 33.9% (n = 20) of them stated that they had their daughter vaccinated against HPV, 49.2% (n = 29) did not have their daughter vaccinated, and 16.9% (n = 10) did not know whether their daughter had been vaccinated or not. In univariate analysis, there was not a significant association between having one's daughter vaccinated and the desire to have one's son vaccinated (p = 0.153), nor between having a daughter vaccinated and the desire to have one's son vaccinated (p = 0.264).

There was also not a significant association between the age of the parents (p = 0.142 and 0.697 for the father and the mother, respectively), the socio-professional category of the parents (p = 0.054), residing in an urban versus a rural area (p = 0.87), the presence of a general practitioner in the town of residence (p = 0.346), and the parents' favourable attitudes toward HPV vaccination.

The preferred contact persons with whom the respondents were most at ease to talk about vaccines against HPV are summarized in Table 5.

For the parents, 59.8% (n = 76) found it easy to discuss sexuality and STIs with their son, 34.6% (n = 44) found this subject difficult but necessary, and 5.6% (n = 7) expressed not having

**Table 5. The preferred contact person with whom the respondents would be most at ease to talk about vaccines against HPV.**

|  | Children (N = 145) | Parents (N = 127) |
|---|---|---|
| **Preferred contact person to talk about the vaccine against HPV** | | |
| Family (the parents for children) | 55.2% (n = 80) | 21.3% (n = 27) |
| Friends | 26.9% (n = 39) | 22.8% (n = 29) |
| Treating physician | 44.1% (n = 64) | 73.2% (n = 93) |
| School doctor or nurse | 29.7% (n = 43) | 28.3% (n = 36) |
| Other doctors | 8.3% (n = 12) | 10.2% (n = 13) |
| Teacher | 7.6% (n = 11) | 4.7% (n = 6) |
| Others | 6.9% (n = 10) | 2.4% (n = 3) |
| **Preferred contact person of the children to talk about STIs** | | |
| Family (the parents for children) | 50.3% (n = 73) | - |
| Friends | 49% (n = 71) | - |
| Treating physician | 24.1% (n = 35) | - |
| School doctor or nurse | 21.4% (n = 31) | - |
| Other doctors | 4.1% (n = 6) | - |
| Teacher | 2.1% (n = 3) | - |
| Others | 8.3% (n = 12) | - |

had an opportunity to talk about it and they deemed the current situation with their son to be "strained" or they even considered it to be a non-productive subject. Of the parents who answered, 63% (n = 80) stated that providing the vaccination to boys would not lead to earlier sexual relationships and/or more at-risk behaviours, 26.8% (n = 34) did not have an opinion, and 10.2% (n = 13) thought that this would result in sexual disinhibition. Lastly, 74% (n = 60) of the parents deemed it important that the sexual partners of their children were vaccinated against HPV, 39.4% (n = 50) did not have an opinion, and for 13.4% (n = 17), this did not seem important to them.

## Discussion

The objective of our study was to evaluate the acceptability of the vaccine against HPV among boys aged 13 to 18 years and their parents. We were able to show that approximately 2/3 of the respondents were either opposed to being vaccinated against HPV or were reluctant to do so.

The acceptability of the HPV vaccine (without prior information in our study) was low among the children and their parents. Nonetheless, only approximately one in ten refused the vaccination from the outset, and the majority were undecided. The rate of acceptability of the children was close to the results of a French study, carried out with male high school students, in which 34.4% of the respondents accepted the HPV vaccination, 24% refused it, and 41% were undecided [17]. Another study, carried out in 2018 with 177 parents of boys aged 11 to 19 years found a rate of acceptability of the vaccination of 41%, while 12% refused it, and 47% were undecided [18]. This low rate was expected: on the one hand, it is in keeping with the low rate of vaccine cover among young girls in France (19.1% in 2016 in France) [19] and, on the other hand, because, in the countries that have implemented universal HPV vaccination several years ago, the vaccination coverage of boys is still lower than that of girls (Australia [20, 21], Italy [22], United States [23]). In comparison, an Italian study showed that although the refusal rate was roughly the same as in France (27.0%), there was much less reluctance (1.4%) [24]. France is thus consolidating its position as a sceptical country when it comes to vaccination [15].

In regard to the parents, a review of the literature published in 2010 reported a higher acceptability of vaccination against HPV in boys than in our study, varying from 68% to 88% [13]. However, this review was carried out in countries such as the United States, Canada, the Netherlands, as well as the United Kingdom, where vaccination coverage of young girls was already higher than in France. More recently, in 2015, a pan-European study reported a parental acceptability for HPV vaccination in boys that varied between 49% and 75% according to the country in question. More extensive refusal by French parents than in our study has been reported (34% refused the vaccination *vs.* 11.8% in our study) and the proportion of those who were undecided was 17% (*vs.* 50.4% for HPVac) [25]. Unlike our study, a brief oral intervention was carried out among the parents before they responded to the questionnaires. The naive nature of our sample may hence have increased the proportion of those who were undecided. A British study performed without prior information already noted that this negatively impacts the parental acceptability [26]. Conversely, the parental acceptability of the anti-HPV was improved when prior information regarding the disease and the direct benefits of the vaccine was provided [25–27].

The factors positively associated with the acceptability of the vaccine were closely linked with knowledge of HPV and particularly with the perception of a risk linked with infection by HPV (among the parents and the children), with the sense of protecting oneself against serious illnesses and genital warts, and with recognition of the efficacy of the vaccine (among the children). The perception of risk linked with infection by HPV is, in fact, one of the factors often correlated with the acceptability [14, 17, 24, 28, 29]. Good knowledge of the virus and of its complications could allow the acceptability of the vaccine against HPV to be increased and hence improve the vaccination coverage.

Unfortunately, we noted that the knowledge of the population of our study was often insufficient. While the mode of transmission of the virus was known by more than ¾ of the children and the parents, this was not the case for 20% of the respondents, and 60% thought that use of a condom was a way to prevent HPV infection. In Italy, where multiple studies have been conducted on HPV, only 55% of the surveyed students reported of having heard about the HPV infection, the majority of whom were aware that it can be sexually transmitted and 56–88% thought that condoms constituted effective protection [24, 30]. Compared to studies carried out overseas, the results of our study are nonetheless reassuring. In Turkey [31], in Malaysia [32], and in the Bahamas [33], the mode of HPV transmission was not known by 57.6%, 58.9%, and 36.4% of men, respectively. In Italy, it has been shown that parents suffer from gaps in their knowledge when it comes to HPV infection and its prevention by means of vaccination, as well as regarding HPV-related diseases [24].

Despite a mostly favourable opinion of vaccination in general (73.8% of the children and 70% of the parents), acceptability of the HPV vaccine (36.6% of the children and 37.8% of the parents) remained low. The mere fact of in principle being in favour of vaccination is hence not enough to commit to the vaccination, and the act of having oneself vaccinated or to having one's son vaccinated against HPV requires access to additional information. As reported in previous publications, the majority of parents have favourable attitudes in regard to adolescent vaccinations in general, with lower levels of support for HPV-specific vaccination [34]. On the other hand, being opposed to vaccination in general was a significant limitation linked with refusal to be vaccinated against HPV, as was the view that the vaccine was of no relevance to one's child. This reflects the parents' lack of information regarding the risk factors for the disease.

A number of studies have, however, highlighted that the risk of infection by HPV tends to be underestimated by both boys [30] and their parents [24, 35]. Fear of altering sexual behaviour and an increase in risk-taking once vaccinated were also a reason cited by approximately

10% of the parents in our study, which was also a reason cited in other foreign studies by parents and sometimes even by doctors [24, 36]. However, after implementing a vaccination program in the school setting, the at-risk sexual behaviours reported by adolescents in Canada decreased or remained the same [37]. Another study, in the USA [38], did not find that there was a change in the sexual behaviours of the youths surveyed (no change in the age at which they had their first sexual relationship, nor an increase in the number of sexual partners).

For the children, the main impediment found was the refusal of the parents (p = 0.003). This notion implies that while children need to be informed of the risks linked with HPV and the existence of a vaccine against this virus, this risks being insufficient if their parents are not in support of the vaccination. This is in line with other studies that claim that knowledge is not a direct predictor of the acceptability of the vaccine [30]. The latter should hence be the primary target for promotion of this vaccination.

Fear of the secondary effects of the vaccine against HPV was cited by a third of the cases by the respondents, although this argument was not significantly associated with the unwillingness to be vaccinated. This factor is encountered frequently in the literature [29], both in children [30] and their parents [39]. The boys tended to express concern that the injection could be painful, while the parents tended to be concerned about the safety of the vaccine. However, these concerns were generally offset by the perceived advantages of the anti-HPV vaccine. In France, fear of the potential secondary effects is a predominant argument that is regularly cited [40].

No correlation between the acceptability of the vaccine and the sociodemographic criteria was found in our study. However, the link between the willingness to be vaccinated and social status has not been clearly defined by previous studies. Some of these have indicated more frequent refusal of the vaccination in populations that are more vulnerable or further away from care [41], and in populations that had less knowledge in regard to sexuality [42]. On the other hand, another study found that parents with a higher level of education and income were less likely to initiate HPV vaccination [34]. This fuzzy correlation between education and vaccine acceptance has also been highlighted in the WHO report on the determinants of vaccine hesitancy [43, 44].

Lastly, our study did not find a significant association between having a daughter vaccinated against HPV and the desire to have one's son(s) vaccinated. These results do not fit with two French studies that found a significant link between having vaccinated one's daughter against HPV and the intention to vaccinate their male siblings [18, 45].

Thus, the lack of information regarding HPV infection, as well as of the existence of a safe vaccine, are major impediments of acceptability of the HPV vaccine. HPV vaccination in France in young girls has been recommended since 2007. However, no large-scale vaccination campaign has been carried out by the Ministry of Health to inform about the risks associated with the virus and the benefits of vaccination. More information, therefore, needs to be provided to children and their parents, the latter most often being the decision-makers. As it turns out, most of the parents in our study (62%) had already been made aware of vaccination against HPV through the media, as also described in the literature [30]. As noted in the WHO report, the media are determinants of the willingness to be vaccinated, as contextual influences [43, 44]. Vaccination hesitancy appears to be more common among parents who have often received non-verified and/or erroneous information through this communication medium [39, 46]. It is, therefore, understandable that some information can be highly detrimental to the acceptability of vaccines, particularly the HPV vaccine [47, 48], and health professionals should play a more prominent role in this regard as they are a trusted and familiar source of information for the individuals concerned. Health professionals, as individual and group

influencers [43] can, in fact, be a positive influence by supplying information and by providing vaccinations [49–53]. Their recommendations have been shown to be strongly positively associated with knowledge, acceptability, and uptake of vaccinations [54, 55], and parents tend to express a high level of trust in doctors to do what is in the best interest of the general public [56].

Nevertheless, it should be kept in mind that they can also adversely affect this when they do not know how to address the patients' reservations or if they advise against using the vaccine [57]. Hence, they need to be properly trained to initiate dialogue, to recommend the vaccine, and to clarify misunderstandings regarding vaccination.

In terms of the school doctor, they only have an advisory and advocacy role, as the vaccination is no longer carried out at schools. However, the countries with the best levels of vaccine coverage are those that provide vaccinations at school. In light of this, in 2014 the French High Council for Public Health declared that vaccination at school would allow a higher percentage of young girls to be reached, independently of their social level [58].

The HPVac study suffers from a number of limitations. The most important one is the low participation rate, which can have multiple causes. First, the multiplicity of the partners involved in getting the questionnaire to the children and then to the parents in question was probably a significant impediment. Furthermore, completion of the questionnaires was on a voluntary basis. Lastly, the sometimes controversial political context surrounding the promotion of vaccination, such as the implementation of citizen consultation and compulsory vaccination of infants, may have limited the participation of the facilities and hence of the parents [59, 60]. This low rate of participation can lead to selection bias, thus potentially affecting the validity of the study and its generalization. Focusing on the socio-demographic characteristics of the respondents, we noted that the proportion of managers and employees was higher than the department data [61], while the intermediate occupations were less represented. Dubé *et al.* observed the same results in their Canadian study [56]. This could have led to the selection of respondents who were the most aware of the prevention of infections linked to HPV and hence an overestimation of the favourable opinion in our sample. Finally, as this work was a cross-sectional study, it is susceptible to biases relating to recruitment, recall, and social acceptability bias. The identified associations identified may be difficult to interpret as it is difficult to draw conclusions regarding the direct causal inferences and the direction of causality. Despite these limitations, this survey provides an important contribution to this topic, as it is the first to report the knowledge and acceptability regarding HPV vaccination in a population of boys and their parents in France.

In conclusion, the results of this study highlight a lack of information among parents and their children about HPV and its vaccination. It also makes it clear that taking time to discuss the consequences of a HPV infection and the relevance of vaccinating boys as well as girls would help convince reluctant parents of the merits of the vaccine. The children then generally go along with their parent's choice. Recommendations in favour of extending the vaccination against HPV to boys were issued in 2019 by the French National Authority for Health [62] and validated by the FNAH for implementation in the 2020 vaccination schedule. There is, therefore, now a need to train health professionals, as well as information campaigns, regarding the risks of infection by HPV and the means for prevention are today necessary.

## Acknowledgments

We would like to thank the boys, parents, and teams from middle and high schools who participated in this study. We would like to thank Sophie Domingues for her work of proofreading and correction.

## Author Contributions

**Conceptualization:** Maëva Lefebvre, Julie Coutherut, Valérie Briend-Godet.

**Investigation:** Antoine Grégoire, Anita Meireles.

**Methodology:** Maëva Lefebvre, Morgane Péré, Julie Coutherut, Valérie Briend-Godet.

**Project administration:** Valérie Briend-Godet.

**Software:** Morgane Péré.

**Supervision:** Charlotte Biron, François Raffi.

**Validation:** Jean-François Huon, Charlotte Biron, François Raffi, Valérie Briend-Godet.

**Writing – original draft:** Jean-François Huon, Valérie Briend-Godet.

**Writing – review & editing:** Jean-François Huon, Valérie Briend-Godet.

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
