## [Decision Letter · Decision Letter 0]

9 Apr 2020

PONE-D-20-03583

Evaluation of the acceptability in France of the vaccine against papillomavirus (HPV) among middle and high school students and their parents: a prospective, multicenter departmental and descriptive survey

PLOS ONE

Dear Dr. Huon,

Thank you for submitting your manuscript to PLOS ONE. After careful consideration, we feel that it has merit but does not fully meet PLOS ONE’s publication criteria as it currently stands. Therefore, we invite you to submit a revised version of the manuscript that addresses the points raised during the review process.

ACADEMIC EDITOR: The title of the manuscript should be modified in Evaluation of the acceptability in France of the vaccine against papillomavirus (HPV) among middle and high school students and their parents 

We would appreciate receiving your revised manuscript by May 10, 2020. To enhance the reproducibility of your results, we recommend that if applicable you deposit your laboratory protocols in protocols.io, where a protocol can be assigned its own identifier (DOI) such that it can be cited independently in the future. For instructions see: http://journals.plos.org/plosone/s/submission-guidelines#loc-laboratory-protocols

We look forward to receiving your revised manuscript.

Kind regards,

Italo Francesco Angelillo, DDS, MPH

Academic Editor

PLOS ONE

2. Please amend your current ethics statement to address the following concerns: Please explain why was written consent was not obtained, how you recorded/documented participant consent, and if the ethics committees/IRBs approved this consent procedure.

Reviewers' comments:

Reviewer's Responses to Questions

**Comments to the Author**

1. Is the manuscript technically sound, and do the data support the conclusions?

Reviewer #1: Yes

Reviewer #2: Partly

2. Has the statistical analysis been performed appropriately and rigorously? 

Reviewer #1: No

Reviewer #2: No

3. Have the authors made all data underlying the findings in their manuscript fully available?

Reviewer #1: Yes

Reviewer #2: Yes

4. Is the manuscript presented in an intelligible fashion and written in standard English?

Reviewer #1: Yes

Reviewer #2: No

5. Review Comments to the Author

Reviewer #1: The manuscript provides an interesting insight on knowledge and acceptance towards HPV vaccination in French male adolescents and their parents. However, some revisions are required.

1. Abstract. “Being discouraged by their parents and parents thinking that their child is not at risk were arguments cited and significantly associated with a willingness to be vaccinated.” Is it correct?

It’s in contrast to what it is stated in the results section, line 101-102: “Conversely, factors impeding HPV vaccination for the children were: being opposed to vaccination in general (p < 0.001) and when it was discouraged by their parents (p=0.003).”

2. Introduction. It would be useful to compare the HPV vaccine schedule in other UE countries in this section, not only in the discussion, to give the reader an idea of the European context.

I would also suggest elaborating on why the “climate of distrust in regard to vaccination in France” still persists.

3. Methods. Why did the authors conduct only univariate analyses without performing logistic regression analyses? It is well known that the associations coming from univariate analyses only describe two quantities, and therefore they are not particularly meaningful to describe relationships.

4. Line 95-126. It is difficult to follow the text going back and forth from children’s responses to parents’ responses. Would it be possible to also synthesize the information in a table, distinguishing parents’ replies from the children’s ones?

5. Line 148-149. “We were able to show that approximately 2/3 of the responders were either opposed or hesitant to the lack of information regarding the virus and the vaccine due to the lack of information regarding the virus and the vaccine”. What is evidence to substantiate this? Line 113-115 report that “A lack of knowledge regarding the vaccine, fear of the secondary effects, and the fact that the subject was not raised by the treating doctor were the most often cited arguments, although they were not significant”.

6. Discussion. What is a possible explanation of the lack of information claimed by the surveyed parents? Is it something consistently reported in France? Or in Europe?

7. Discussion. To more substantiate the background for these results, I would suggest including references to other research conducted on vaccine hesitancy, in particular the work of WHO SAGE Working Group on Vaccine hesitancy, and other studies dealing with the same topic. Relevant papers may include the following:

- Napolitano F, D’Alessandro A, Angelillo IF. Investigating Italian parents’ vaccine hesitancy: A cross-sectional survey. Hum Vaccin Immunother 2018;1–8.

- MacDonald NE, the SAGE Working Group on Vaccine Hesitancy. Vaccine hesitancy: Definition, scope and determinants. Vaccine 2015;33:4161–4.

- Napolitano F, Napolitano P, Liguori G, Angelillo IF. Human papillomavirus infection and vaccination: Knowledge and attitudes among young males in Italy. Hum Vaccin Immunother. 2016;12(6):1504-10.

- Dubé E, Gagnon D, Ouakki M, Bettinger JA, Witteman HO, MacDonald S, et al. Measuring vaccine acceptance among Canadian parents: A survey of the Canadian Immunization Research Network. Vaccine 2018;36:545–52.

8. Line 160-163. Please be careful not to repeat the results in the discussion section.

9. Line 269-278. These conclusions do not relate to the findings of this study. Consider perhaps moving these concepts to the discussion part and replacing the conclusion with an appropriate synthesis of the results.

Reviewer #2: The authors present the results of a survey to assess the acceptability of HPV vaccination in men, among middle and high school students and their parents in the Loire-Atlantique department, France. Understanding the attitudes and perceived efficacy regarding HPV vaccination in men is pivotal to develop and provide effective programs and to establish health policies to increase the awareness of the risks associated with HPV infection and promote the vaccination.

I find the topic of this study is of potential interest to the international scientific community, but it has major pitfalls that have to be overcome before the acceptance.

I think the main issue of the paper is the extremely low participation rate (1.4% for parents and 1.6% for students), a major limitation that undermines the validity of this study and suggests that the obtained results could not be representative of a wider population.

Abstract

It would be better understood if the summary was shown in a structured way (Background, Methods, Results, Conclusion).

Please, add the study period in the methods.

Introduction

Page 3, line 1, use the abbreviation of HPV for the first time and then throughout the manuscript (e.g. line 3, line 7, etc.).

Methods

I suggest to delete the paragraph: “The target populations of the study were comprised of grade 4 and grade 3 middle school and high school students attending schools in the Loire-Atlantique department (HPVac_child) and their parents (HPVac_parent).”. It is redundand. I suggest to add the school grade (4 and 3) in the first paragraph.

I suggest to move and to incorporate the paragraph about the study objectives at the end of the introduction.

Page 5, lines 16 and 17: I suggest to delete the criteria for non-inclusion.

The authors should mention whether or not an incentive was offered for completion of the questionnaire.

Were study participants assured for confidentiality? Please, clarify.

The survey questionnaire was validated by a pilot study. It is not given any information regarding any modifications in the questionnaire after the pre-testing study. If any, what was the extent of the modifications?

The question remains as to the selection and representativeness of the population in this study. The selection bias would have affected the external validity in terms of generalizing the findings to the wider population, and I am not sure if the attempt of the authors to minimize the selection bias was enough. There might be a large generalizability problem.

All of the abovementioned issues suggest serious study limitations that should be addressed in the Limitations section.

It would be helpful to have a copy of the questionnaire that was used in the appendix.

Statistical analysis

The statistical analysis is not adequate and incomplete. It would be very useful to generate and present the results of a logistic regression analysis, since it would give more robustness to the results.

Discussion

I believe the Authors have posed an effort to adequately characterize the acceptability of HPV vaccines among middle and high school students and their parents in France. But, as I mentioned earlier, it would be also interesting exploration of the potential associated factors through a multivariate data analysis techniques.

I suggest to expand the literature review to generate a better structured discussion. The following papers have to be cited and commented: Hum Vaccin Immunother. 2014;10(9):2536-42, Hum Vaccin Immunother. 2016 Jun 2;12(6):1504-10, Hum Vaccin Immunother. 2016;12(1):47-51.

I suggest to add a comment about potential determinants of vaccine hesitancy, e.g. according to the WHO Strategic Advisory Group of Experts (SAGE) on Immunization. The following studies have to be cited and commented: Bianco et al. Vaccine 2019;37:984-999 and Napolitano et al. Hum Vaccin Immunother. 2018 Jul 3;14(7):1558-1565.

The study findings showed that the treating physicians and the school doctor were the two most often cited sources of information. I suggest to discuss more thoroughly the pivotal role of healthcare providers in promoting HPV vaccination in different setting and in specific risk groups. I suggest to cite and comment other studies (e.g. Napolitano F, et al. PLoS One. 2018 Mar 29;13(3):e0194920; Landis K et al. Vaccine. 2018 Jun 7;36(24):3498-3504; D’Alessandro et al. Hum Vaccin Immunother 2018;14:1573-1579; etc..).

In the limitations of the study, the authors did not mention the main limits of the cross-sectional survey.

I suggest you have a fluent, preferably native, English-language speaker thoroughly copyedit your manuscript for language usage, spelling, and grammar.

6. PLOS authors have the option to publish the peer review history of their article (what does this mean?). If published, this will include your full peer review and any attached files.

Reviewer #1: No

Reviewer #2: No

---

## [Author Response · Author response to Decision Letter 0]

7 May 2020

ACADEMIC EDITOR: 

The title of the manuscript should be modified in Evaluation of the acceptability in France of the vaccine against papillomavirus (HPV) among middle and high school students and their parents 

The authors thank the Academic Editor for this comment. This change has been made.

We also inform the editor of the addition among the authors of Morgane Péré who has contributed in a very important way to the methodology and the multiple necessary statistical calculations.

REVIEWERS:

Reviewer #1: 

Q1. Abstract. “Being discouraged by their parents and parents thinking that their child is not at risk were arguments cited and significantly associated with a willingness to be vaccinated.” Is it correct?

It’s in contrast to what it is stated in the results section, line 101-102: “Conversely, factors impeding HPV vaccination for the children were: being opposed to vaccination in general (p < 0.001) and when it was discouraged by their parents (p=0.003).”

R1. The authors thank Reviewer 1 for his remark. We apologize for this confusion. 

Although all the statements were true, as described in the table (see R5), the results highlighted in the Abstract and in the Results section were not the same. This discrepancy has been corrected.

Q2. Introduction. It would be useful to compare the HPV vaccine schedule in other UE countries in this section, not only in the discussion, to give the reader an idea of the European context. 

R2. Following the Reviewer’s comment, this point has now been added to the “Introduction” section:

“Other countries have already extended their vaccination guidelines to include boys. For example, 15 countries in Europe have included the vaccination of boys in their immunization schedules (Austria, Belgium, Croatia, the Czech Republic, Denmark, Germany, Ireland, Italy, Liechtenstein, Luxembourg, Norway, Slovakia, Sweden, Switzerland, and the United Kingdom)”.

Q3. I would also suggest elaborating on why the “climate of distrust in regard to vaccination in France” still persists.

R3. In light of the Reviewer’s comment, we have expanded on this point in the “Introduction” section: 

“Fear of side effects, misinformation spread by anti-vaccine lobbying, and poor knowledge of vaccine-preventable diseases is exacerbated by a loss of confidence in institutions. Indeed, there has been a succession of health scandals (or that are perceived as such), in recent years in France. Among these, the controversy surrounding the side effects of the hepatitis B vaccine, which led to the suspension of the school vaccination campaign at the end of 1998, could explain this lack of confidence”.

Q4. Methods. Why did the authors conduct only univariate analyses without performing logistic regression analyses? It is well known that the associations coming from univariate analyses only describe two quantities, and therefore they are not particularly meaningful to describe relationships.

R4. As requested by the two Reviewers, we have performed additional statistical analyses. Thus, we have performed logistic regression tests, which we have incorporated into the Results section.

Q5. Line 95-126. It is difficult to follow the text going back and forth from children’s responses to parents’ responses. Would it be possible to also synthesize the information in a table, distinguishing parents’ replies from the children’s ones?

R5. In accordance with the Reviewer’s comment, and to avoid confusion, we have summarized the information in a table (Table 4).

Q6. Line 148-149. “We were able to show that approximately 2/3 of the responders were either opposed or hesitant to the lack of information regarding the virus and the vaccine due to the lack of information regarding the virus and the vaccine”. What is evidence to substantiate this? Line 113-115 report that “A lack of knowledge regarding the vaccine, fear of the secondary effects, and the fact that the subject was not raised by the treating doctor were the most often cited arguments, although they were not significant”.

R6. The authors thank the Reviewer for this comment. 

We agree that our method does not reveal a causal link between the lack of information and the reluctance to be vaccinated. We have, therefore, altered the sentence accordingly by deletion of this assertion. However, the parents and the children made it clear that they are not sufficiently informed, and our analyses show a trend: the probability of responding negatively to vaccination was higher when the parents stated that they did not know enough about the vaccine.

Q7. Discussion. What is a possible explanation of the lack of information claimed by the surveyed parents? Is it something consistently reported in France? Or in Europe?

R7. The authors would like to thank the Reviewer for their helpful comments. Although vaccination against HPV has been recommended in France since 2007, no public health campaign has been conducted in France to inform the population. The authors have, therefore, added a paragraph on this subject.

Q8. Discussion. To more substantiate the background for these results, I would suggest including references to other research conducted on vaccine hesitancy, in particular the work of WHO SAGE Working Group on Vaccine hesitancy and other studies dealing with the same topic. Relevant papers may include the following:

- Napolitano F, D’Alessandro A, Angelillo IF. Investigating Italian parents’ vaccine hesitancy: A cross-sectional survey. Hum Vaccin Immunother 2018;1–8. 

- MacDonald NE, the SAGE Working Group on Vaccine Hesitancy. Vaccine hesitancy: Definition, scope and determinants. Vaccine 2015;33:4161–4. 

- Napolitano F, Napolitano P, Liguori G, Angelillo IF. Human papillomavirus infection and vaccination: Knowledge and attitudes among young males in Italy. Hum Vaccin Immunother. 2016;12(6):1504-10. 

- Dubé E, Gagnon D, Ouakki M, Bettinger JA, Witteman HO, MacDonald S, et al. Measuring vaccine acceptance among Canadian parents: A survey of the Canadian Immunization Research Network. Vaccine 2018;36:545–52.

R8. These references have been incorporated into the Discussion section, which has been amended accordingly. These changes are highlighted in the manuscript.

Q9. Line 160-163. Please be careful not to repeat the results in the discussion section.

R9. The authors thank Reviewer 1 for his remark. The manuscript has been updated accordingly.

Q10. Line 269-278. These conclusions do not relate to the findings of this study. Consider perhaps moving these concepts to the discussion part and replacing the conclusion with an appropriate synthesis of the results.

R10. The authors agree with the Reviewer’s comment. The conclusion has been changed entirely. It has been replaced with a summary of the study and it has been integrated at the end of the Discussion section.

 

Reviewer #2:

Q11. Abstract. It would be better understood if the summary was shown in a structured way (Background, Methods, Results, Conclusion). Please, add the study period in the methods.

R11. The authors fully agree with the Reviewer’s comment. The “Discussion” section has been modified as recommended. A structured summary has been added, as has the study period.

Q12. Introduction. Page 3, line 1, use the abbreviation of HPV for the first time and then throughout the manuscript (e.g. line 3, line 7, etc.).

R12. The authors thank Reviewer 2 for this particularly relevant remark. After having explained the abbreviation for the first time at line 1 of the Introduction, it was then used throughout the manuscript, instead of the term papillomavirus.

Q13. Methods. I suggest to delete the paragraph: “The target populations of the study were comprised of grade 4 and grade 3 middle school and high school students attending schools in the Loire-Atlantique department (HPVac_child) and their parents (HPVac_parent).”. It is redundand. I suggest to add the school grade (4 and 3) in the first paragraph.

R13. The authors have altered the manuscript accordingly.

Q14. I suggest to move and to incorporate the paragraph about the study objectives at the end of the introduction.

R14. The authors have corrected the manuscript accordingly. The objectives have been moved to the end of the Introduction section.

Q15. Page 5, lines 16 and 17: I suggest to delete the criteria for non-inclusion.

R15. As recommended by the Reviewer, the criteria for non-inclusion have been deleted according to the reviewer’s request.

Q16. The authors should mention whether or not an incentive was offered for completion of the questionnaire.

R16. No incentives were offered to motivate the respondents to complete the questionnaire. This is now specified in Materials and Methods section.

Q17. Were study participants assured for confidentiality? Please, clarify.

R17. Each participant had remote access to the questionnaire through a web link on the mail received. No nominative data was collected, ensuring anonymity and confidentiality of the data. This is now specified in Materials and Methods section.

Q18. The survey questionnaire was validated by a pilot study. It is not given any information regarding any modifications in the questionnaire after the pre-testing study. If any, what was the extent of the modifications?

R18. We thank the Reviewer for his constructive comment. We added further information in the Materials and Methods section some extra information about the modifications of the questionnaire:

“Finally, some questions were deleted due to misunderstanding or duplication. This was the case for questions regarding the notion of increased risk in case of vaccination, the importance of vaccinating partners, or -when a boy respondent had a sister- her vaccination status”.

Q19. The question remains as to the selection and representativeness of the population in this study. The selection bias would have affected the external validity in terms of generalizing the findings to the wider population, and I am not sure if the attempt of the authors to minimize the selection bias was enough. There might be a large generalizability problem.

All of the abovementioned issues suggest serious study limitations that should be addressed in the Limitations section.

R19. The authors fully agree with this comment. We recognize that the low participation rate could potentially lead to a selection bias and thus compromise the validity of the study. As recommended, we have, therefore, added a paragraph about possible bias in the limitations section, which reads as follows:

“The HPVac study suffers from a number of limitations. The most important one is the low participation rate, which can have multiple causes. (…) This low rate of participation can lead to selection bias, thus potentially affecting the validity of the study and its generalization. Focusing on the socio-demographic characteristics of the respondents, we noted that the proportion of managers and employees was higher than the department data, while the intermediate occupations were less represented. Dubé et al. observed the same results in their Canadian study (56). This could have led to the selection of respondents who were the most aware of the prevention of infections linked to HPV and hence an overestimation of the favourable opinion in our sample”.

Q20. It would be helpful to have a copy of the questionnaire that was used in the appendix.

R20. In light of the Reviewer’s comment, the questionnaire has now been added to the “Appendix” section.

Q21. Statistical analysis. The statistical analysis is not adequate and incomplete. It would be very useful to generate and present the results of a logistic regression analysis, since it would give more robustness to the results. It would be interesting exploration of the potential associated factors through a multivariate data analysis technique.

R21. As requested by the two reviewers, we have performed additional statistical analyses. We have performed logistic regression tests and multivariate data analysis which have been incorporated into the Results section.

Q22. Discussion. I suggest to expand the literature review to generate a better structured discussion. The following papers have to be cited and commented: Hum Vaccin Immunother. 2014;10(9):2536-42, Hum Vaccin Immunother. 2016 Jun 2;12(6):1504-10, Hum Vaccin Immunother. 2016;12(1):47-51.

I suggest to add a comment about potential determinants of vaccine hesitancy, e.g. according to the WHO Strategic Advisory Group of Experts (SAGE) on Immunization. The following studies have to be cited and commented: Bianco et al. Vaccine 2019;37:984-999 and Napolitano et al. Hum Vaccin Immunother. 2018 Jul 3;14(7):1558-1565.

The study findings showed that the treating physicians and the school doctor were the two most often cited sources of information. I suggest to discuss more thoroughly the pivotal role of healthcare providers in promoting HPV vaccination in different setting and in specific risk groups. I suggest to cite and comment other studies (e.g. Napolitano F, et al. PLoS One. 2018 Mar 29;13(3):e0194920; Landis K et al. Vaccine. 2018 Jun 7;36(24):3498-3504; D’Alessandro et al. Hum Vaccin Immunother 2018;14:1573-1579; etc..).

R.22 These references have been incorporated into the Discussion section, which has been amended accordingly. The changes are highlighted in the manuscript.

The Landis et al study has not been included in the Discussion section because- on reading the publication- it turned out that most of the work was in regard to the link between vaccination and ethnicity, which we considered to be of little relevance to our study.

Q23. In the limitations of the study, the authors did not mention the main limits of the cross-sectional survey.

R23. The authors thank Reviewer 2 for highlighting this point. A paragraph about the limitations of the cross-sectional studies has been added:

“As this work was a cross-sectional study, it is susceptible to biases related to recruitment, recall, and social acceptability bias. The identified associations may be difficult to interpret as it is difficult to draw conclusions regarding the direct causal inferences and the direction of causality”.

Q24. I suggest you have a fluent, preferably native, English-language speaker thoroughly copyedit your manuscript for language usage, spelling, and grammar.

R24. In light of this request, the manuscript has undergone further copy editing by a certified medical translator.

---

## [Decision Letter · Decision Letter 1]

18 Aug 2020

Evaluation of the acceptability in France of the vaccine against papillomavirus (HPV) among middle and high school students and their parents

PONE-D-20-03583R1

Dear Dr. Jean-Francois,

We’re pleased to inform you that your manuscript has been judged scientifically suitable for publication and will be formally accepted for publication once it meets all outstanding technical requirements.

Kind regards,

Nelly Rwamba Mugo, M.D

Academic Editor

PLOS ONE

Additional Editor Comments (optional):

Reviewers' comments:

Reviewer's Responses to Questions

**Comments to the Author**

1. If the authors have adequately addressed your comments raised in a previous round of review and you feel that this manuscript is now acceptable for publication, you may indicate that here to bypass the “Comments to the Author” section, enter your conflict of interest statement in the “Confidential to Editor” section, and submit your "Accept" recommendation.

Reviewer #1: All comments have been addressed

Reviewer #2: All comments have been addressed

2. Is the manuscript technically sound, and do the data support the conclusions?

Reviewer #1: Yes

Reviewer #2: Yes

3. Has the statistical analysis been performed appropriately and rigorously? 

Reviewer #1: Yes

Reviewer #2: Yes

4. Have the authors made all data underlying the findings in their manuscript fully available?

Reviewer #1: Yes

Reviewer #2: Yes

5. Is the manuscript presented in an intelligible fashion and written in standard English?

Reviewer #1: Yes

Reviewer #2: Yes

6. Review Comments to the Author

Reviewer #1: The manuscript is now suitable for publication. All the reviewers' requests have been addressed. No further revision is needed.

Reviewer #2: (No Response)

7. PLOS authors have the option to publish the peer review history of their article (what does this mean?). If published, this will include your full peer review and any attached files.

Reviewer #1: No

Reviewer #2: No

---

## [Editor Report · Acceptance letter]

25 Aug 2020

PONE-D-20-03583R1 

Evaluation of the acceptability in France of the vaccine against papillomavirus (HPV) among middle and high school students and their parents 

Dear Dr. HUON:

I'm pleased to inform you that your manuscript has been deemed suitable for publication in PLOS ONE. Congratulations! Your manuscript is now with our production department. 

Kind regards, 

on behalf of

Dr. Nelly Rwamba Mugo 

Academic Editor

PLOS ONE